# Hairy Root Induction of *Taxus baccata* L. by Natural Transformation with *Rhizobium rhizogenes*

**Junou He \***, **João Paulo Alves Plácido** [ID], **Irini Pateraki**, **Sotirios Kampranis** [ID], **Bruno Trevenzoli Favero** [ID] **and Henrik Lütken**

Department of Plant and Environmental Sciences, University of Copenhagen, Højbakkegård Allé 9-13, 2630 Taastrup, Denmark
**\*** Correspondence: junou@plen.ku.dk

**Abstract:** Paclitaxel (Taxol®) is a potent anticancer agent, but the widespread pharmaceutical use of paclitaxel is hampered by its limited availability due to low accumulation levels in the native yew (*Taxus* spp.) plants. Currently, hairy root culture is an emerging biotechnological tool that presents several advantages such as reduced costs and higher specialized metabolite production, therefore, its application to paclitaxel production can be of commercial and medicinal interest. The objective of present study was to induce hairy root in *Taxus baccata* L. by transformation with the wild type *Rhizobium rhizogenes* A4 strain. Thus, *T. baccata* was inoculated by three different inoculation methods: (a) ex vitro seedlings inoculation by direct injection of a liquid bacterial culture; (b) ex vitro needles inoculation by liquid co-culturing with bacteria; (c) ex vitro shoots inoculation by dipping liquid bacterial culture. Hairy roots were formed only from ex vitro seedlings inoculated by the direct inoculation method, with transformation efficiency of 14.3%. Formation of hairy roots was observed two months after inoculation. This project forms the basis for the establishment of hairy root cultures from *T. baccata* for the production of paclitaxel.

**Keywords:** ex vitro; seedling; paclitaxel; transformation efficiency





## 1. Introduction

*Taxus baccata* L. is the most commonly cultivated species of *Taxus* in Europe, mainly used for ornamental purposes [1]. The medicinal interest in *Taxus* was low until the potential anticancer properties of paclitaxel from *Taxus* was found [1–4]. Paclitaxel (also called Taxol®) is a tricyclic diterpenoid with the formula $C_{47}H_{51}NO_{14}$ [5,6]. It works as an anticancer agent by binding to tubulin and promoting the assembly of cellular microtubules, and inhibiting their disassembly [7–10]. Paclitaxel was first isolated from the bark extracted from *T. brevifolia* [2]. However, the yields were extremely low at around 0.01% on a dry weight basis [11]. Paclitaxel is also present in other tissues from different *Taxus* species, although the yields are even lower than those of the bark from *T. baccata* [12,13]. The extraction of paclitaxel directly from the harvested trees remains financially and ecologically unsustainable. Therefore, chemists took the challenge of chemically synthesizing paclitaxel, which was first conducted by two different groups in 1994 [14–17]. However, the complexity of the molecule, and the several steps required for its synthesis, led to the process having a low yield, and hence also being financially unsustainable [18]. The discovery of 10-deacetylbaccatin (10-DAB) in 1986 opened the door to the semi-synthesis of paclitaxel [19–22]. Nevertheless, the harvesting and extraction process of 10-DAB is still laborious [23].

Recently, attention has been rapidly moving toward the application of hairy root cultures (HRCs) for the production of many valuable secondary metabolites [24]. Hairy roots (HRs) resulting from natural transformation (i.e., transformation with unmodified bacterial strains of e.g., *Rhizobium rhizogenes* (previously known as *Agrobacterium rhizogenes*)

without the use of recombinant DNA technologies) are not recognized as a genetically modified organism (GMO) in EU and Japan [25–29]. Morphologically, many small, highly branched roots protrude directly from the infection site in a range of plant species, a phenomenon that gave rise to the term hairy root [30]. *R. rhizogenes* contain a root-inducing (Ri) plasmid [31,32]. The pathogenicity of the bacterium is caused by a horizontal gene transfer event, where a portion of the Ri plasmid, the transfer DNA (T-DNA), is transferred and integrated in the genome of the host plant [33,34]. Several genes are present on the T-DNA and at least 18 open reading frames (ORFs) have been described. The most well-characterized genes are the *root oncogenic loci* (*rol*) genes (i.e., *rol*A, *rol*B, *rol*C, and *rol*D), which coincides with ORF 10, 11, 12, and 15, respectively [35,36], which are involved in the initiation of HR formation [37,38].

Four Ri plasmid types have been described as of now: agropine (A4 strain type), cucumopine (NCPPB2659 strain type), mannopine (NCIB8196 strain type), and mikimopine (MAFF301724 strain type) [39]. Unlike the other Ri plasmid types, the agropine type Ri plasmids have a split T-DNA, which is divided into a left T-DNA ($T_L$-DNA) and right T-DNA ($T_R$-DNA) [40]. The $T_L$-DNA contains the *rol* genes and other ORFs [41]. On the other hand, the $T_R$-DNA contains *aux*1 and *aux*2, which are genes responsible for auxin biosynthesis [42,43]. Aside from the T-DNA region, the Ri plasmids also contain virulence (*vir*) genes necessary for the horizontal gene transfer event [31,44].

Secondary metabolite production in HRCs usually have similar or higher yields than the wild type tissues from the same plant [45]. As a result, research on HRCs from *Taxus* spp. has been pursued to increase the production of paclitaxel. Palut-Carcasson [46] tested five strains of *R. rhizogenes* and found that the transformation efficiency of *Taxus brevifolia*. was extremely low, at around 0.5%, and only three HR lines were established, all from the ATCC39207 strain. Huang et al. [47] reported that successful transformation only occurred from shoot explants of *T. brevifolia* inoculated with the ATCC31798 strain of *R. rhizogenes*, with 30% transformation efficiency. Furmanowa and Syklowska-Baranek [48] stated that *T. x media* var. *hicksii* was successfully transformed with *R. rhizogenes* strain LBA9402, but only one HR line was established. Kim et al. [45] inoculated *T. cuspidata*, but no HR formed from the seedlings infected with the A4 strain, whilst the seedlings infected with ATCC15834 and R1000 successfully formed HRs. Spjut [49] found that *T. sumatrana* transformed with the AR1600 and ATCC15834 strains of *R. rhizogenes* yielded a transformation efficiency of 15% and 19%, respectively. A recent report by Sahai and Sinha [50] demonstrated the successful transformation of *T. wallichiana* by using the *R. rhizogenes* MTCC532 strain with an average transformation efficiency of 24%.

In general, the above research proved the prospects of *Taxus* spp. producing HRs through transformation with *R. rhizogenes*, although *Taxus* spp. seems to be somewhat recalcitrant to transformation because of the low transformation efficiency. In the experiment by Furmanowa and Syklowska-Baranek [48], the induced HRs were exposed to methyl jasmonate, and subsequently the yield of paclitaxel increased by 3-fold compared to the paclitaxel contents of bark harvested from *Taxus* spp. Therefore, the production of HRs is a highly promising strategy to boost paclitaxel production. As previously mentioned, HRs have been produced using different *Taxus* species as natural transformation material. However, no HR has yet been obtained from *T. baccata*, the most common wild species in Europe. Furthermore, HRs have not been induced in *Taxus* spp. By the *R. rhizogenes* A4 strain according to current knowledge. Therefore, different *T. baccata* plant materials were used for natural transformation with the A4 strain in the present work, with the aim of producing HRs. In addition, three different bacterial inoculation methods were tested in this study: (a) a direct inoculation ex vitro method; (b) a liquid co-culture ex vitro method; and (c) a dipping method. The importance of this research is that it is the first time to successfully produce HRs using *T. baccata* as a natural transformation material, which can close the knowledge gap. Moreover, the A4 strain was used to inoculate explants in this experiment. The successful generation of HR in the ex vitro method was achieved. This

breakthrough research lays the basis for establishing a platform for generating HRCs from *T. baccata*.

## 2. Materials and Methods

### 2.1. Experimental Design

In this study, five different strategies of bacteria inoculation were used for different *T. baccata* tissues, three of them ex vitro, as follows:

- Direct inoculation ex vitro: ex vitro seedling inoculation by direct injection of a liquid bacterial culture.
- Liquid co-culture ex vitro: ex vitro needle inoculation by liquid co-culturing with bacteria.
- Dipping method: ex vitro shoot inoculation by dipping liquid bacterial culture.
- Direct inoculation in vitro: in vitro shoot inoculation by directly smearing in a solid bacterial culture (Supplementary File S1).
- Liquid co-culture in vitro: in vitro seedling inoculation by liquid co-culturing with bacteria (Supplementary File S2).

### 2.2. Plant Material

Regarding the direct inoculation ex vitro, 10 ex vitro *T. baccata* seedlings (Figure 1A) were obtained from the domesticated material and kept moist until the beginning of the experiment. With respect to the liquid co-culture ex vitro and dipping method, ex vitro *T. baccata* branches (Figure 1B) were obtained from the wild. For the dipping method, young shoots with 6.0 to 10.0 cm in height, and 7.0 to 11.0 cm in width from the branches were selected (Figure 1C). The needles at the branch base, ca. 3.0 cm, were discarded (Figure 1D). For liquid co-culture ex vitro, needles from young shoots that showed no signs of damage or contamination were selected (Figure 1E,F).

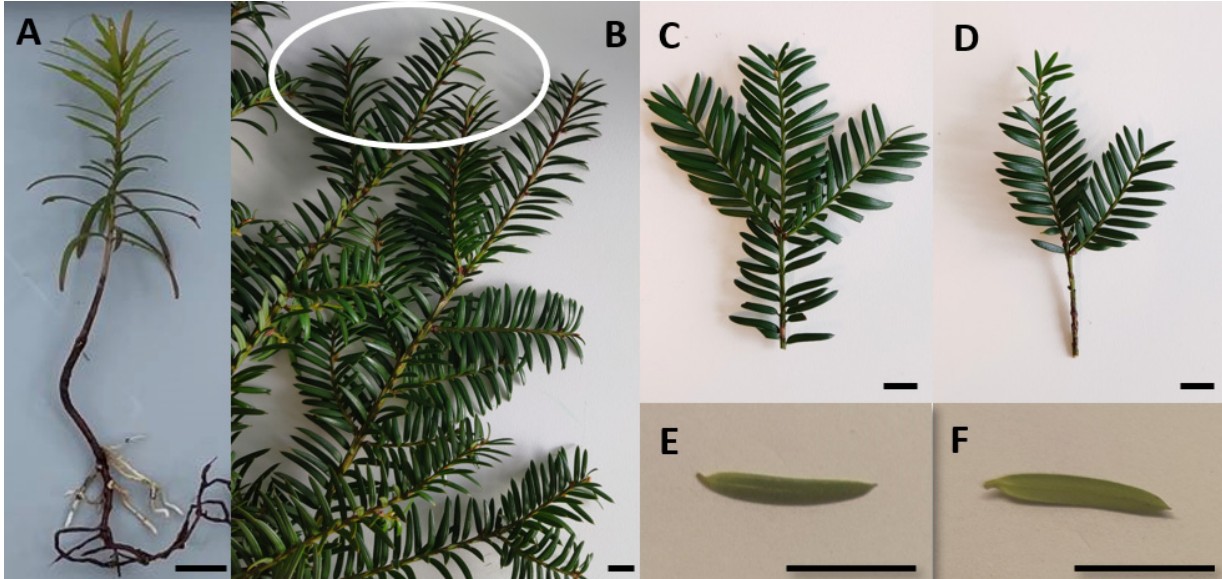

**Figure 1.** Plant material obtained from wild *T. baccata* seedlings and domesticated trees. (**A**) *T. baccata* seedling collected from a garden; (**B**) *T. baccata* branch collected from a garden. The white circle surrounding the sample of plant material selected for the preparation of both the shoot and needles. (**C**) Specimen of a young shoot explant. (**D**) Specimen of a young shoot explant after discarding the needles from approximately 3 cm from shoot base. (**E**) Adaxial side of a needle selected as an explant. (**F**) Abaxial side of a needle selected as an explant. Both sides of the needle were green and free of contamination. Bars = 1 cm.

### 2.3. Bacterial Strain

The A4 strain (from Dr. David Tepfer, Biologie de la Rhizosphère, INRA, Versailles, France) of *Rhizobium rhizogenes* was used for the induction of HRs. A total of 1 mL glycerol stock of the bacteria was cultured in 9 mL of liquid malt yeast agar (MYA) medium [51] and shaken at 200 rpm for 24 h at 28 °C. The culture was then diluted in 90 mL of MYA medium and incubated again at 28 °C for 24 h in a shaker (Innova 4430, Mississauga, Canada) at 200 rpm. After that, the optical density (OD) of the bacterial culture was adjusted to $OD_{600} = 0.5$ with MYA medium, and 0.67 mL $L^{-1}$ 1-(4-hydroxy-3,5-dimethoxyphenyl)-ethanone (acetosyringone) (Duchefa, Haarlem, The Netherlands) was added to the culture. Then, the bacterial culture was shaken at 28 °C, 200 rpm, for an additional 4 h. Finally, the bacterial culture was diluted to $OD_{600} = 0.5$ with the MYA medium.

### 2.4. Ex Vitro Seedlings Inoculation by Direct Injection of a Liquid Bacterial Culture

The ex vitro seedlings were inoculated using an adapted protocol from Nguyen and Searle [52]. *T. baccata* seedlings were placed in a 13.0 × 10.0 cm pot (Billund Potter, Billund, Denmark) containing peat, and a top 2.0 cm layer of vermiculite (Figure 2). The seven seedlings (n = 7) were inoculated near the roots by injecting 0.05 mL of liquid bacterial culture ($OD_{600} = 0.5$) with a 1 mL sterile syringe (BD Plastipak, San Agustin, Spain), using a 27 G × $\frac{3}{4}$″ (0.40 × 20 mm) sterile syringe needle (Braun, Melsungen, Germany), while the remaining three seedlings were inoculated with MYA medium as the control. Each seedling was inoculated twice. The inoculated region was covered by the vermiculite layer. The seedlings were grown under greenhouse conditions at 21 °C for 2 months, after which they were taken out of the pots to check for HR formation.

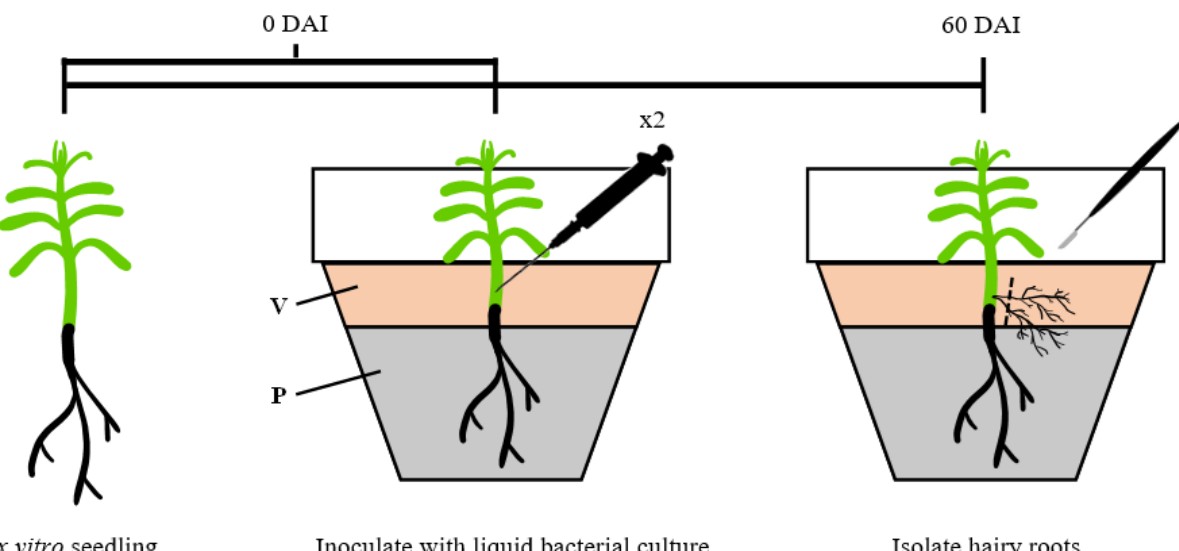

**Figure 2.** Schematic representation of the inoculation of ex vitro seedlings by the direct inoculation method. DAI: days after inoculation; V: vermiculite; P: peat.

### 2.5. Ex Vitro Needle Inoculation by Liquid Co-Culturing with Bacteria

The needle explants were washed with sterile water containing 18.7 mL $L^{-1}$ soap (Citop Zero, Greenspeed, Ryswick, The Netherlands) for 30 min in a shaker at 100 rpm, followed by 70% (*v/v*) ethanol (VWR, Rosny-sous-Bois, France) for 5 min, bleaching with 10.0 g $L^{-1}$ calcium hypochlorite (Sigma-Aldrich, Munich, Germany) supplemented with three drops of Tween 20 (Sigma-Aldrich, Munich, Germany) for 50 min, and finally, sterile water with 100.0 mg $L^{-1}$ nystatin (Duchefa, Haarlem, The Netherlands) for 30 min. Finally, the needles were rinsed three times with sterile water.

After surface sterilization, needles were wounded on their abaxial side with a scalpel (Sigma-Aldrich, Munich, Germany) (Figure 3). Fifteen needles were placed in a 15 mL Falcon tube (Nerbe Plus, Harburg, Germany). Based on the research from Hiei et al. [53] and

Khanna et al. [54], the needles were subjected to a heat treatment in a water bath at 45 °C for 10 min, followed by a quick cooling period of 1 min on ice. A total of 15 mL of liquid bacterial culture was added to each Falcon tube, vortexed, and incubated for 5 min at room temperature. The needles were then subjected to centrifugation at 4500 rpm for 10 min at 25 °C. A second incubation period of 20 min at room temperature followed. Then, the needles were transferred to the co-cultivation media—$\frac{1}{2}$ Murashige & Skoog (MS) media [55] supplemented with 15 mg L$^{-1}$ acetosyringone—for two days, with the adaxial side facing the medium. The same procedure was performed with needles and pure MYA as the control. Thirty repetitions for each treatment (n = 30) with 15 needles per repetition, totaling 900 explants were used. After co-cultivation, they were transferred in tubes to wash with sterile water containing 100 mg L$^{-1}$ of ticarcillin disodium/clavulanate potassium (mixed in ratio 15:1, Timentin) (Duchefa, Haarlem, The Netherlands). Then, the needles were transferred to the rooting media ($\frac{1}{2}$ MS + 200 mg L$^{-1}$ timentin + 5 g L$^{-1}$ activated charcoal (Duchefa, Haarlem, The Netherlands)). The needles were always kept under dark conditions at 25 °C and changed to fresh medium every 2 weeks.

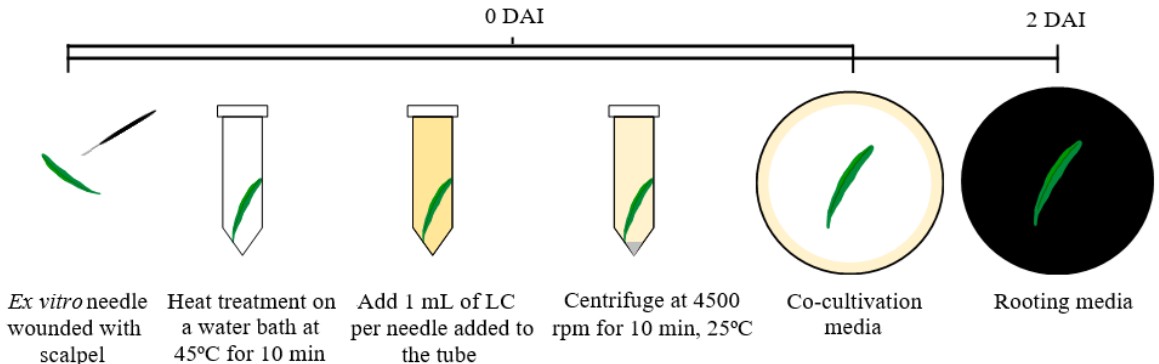

**Figure 3.** Schematic representation of the inoculation of ex vitro needles by the liquid co-culture method. DAI: days after infection; LC: liquid culture of *R. rhizogenes*.

*2.6. Ex Vitro Shoot Inoculation by Dipping Liquid Bacterial Culture*

The base of young shoot explants was cut and the shoots were placed in 1.5 mL tubes containing 1 mL of liquid bacterial culture, and were left overnight under dark conditions at room temperature (Figure 4). Twenty shoots (n = 20) were utilized for the inoculation, and the same number of shoots for the controls were placed under the same conditions, but in contact with the MYA medium. After inoculation, the explants were transferred to a 13.0 × 10.0 cm pot containing peat and grown under the same greenhouse conditions.

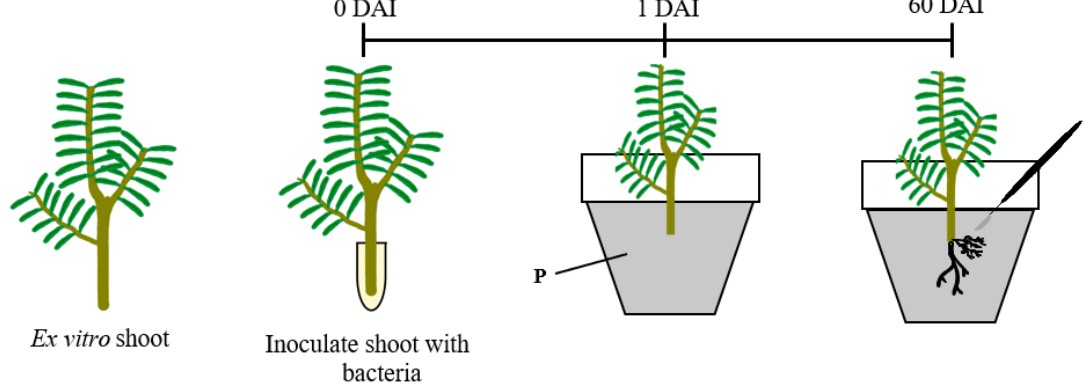

**Figure 4.** Schematic representation of the inoculation of ex vitro shoots by the dipping method. DAI: days after infection.

*2.7. Molecular Analysis for the Confirmation of Transgenic HRs*

Up to 100 mg of putatively transformed root was collected to a 1.5 mL tube and stored at $-80\ °C$ until the DNA was extracted. The DNA extraction was performed following the instructions of the Plant DNA Isolation Kit (Takara, Kusatsu, Japan). DNA concentration was measured on a Nano-drop (ND-1000, ThermoFisher Scientific, MA, USA). Subsequently, each sample was adjusted to the DNA concentration of 60 ng $\mu L^{-1}$.

The presence of *rol*D was used to confirm the T-DNA insertion, while 18S was the internal control for DNA integrity and the presence of *vir*D was used to check for the remnants of *R. rhizogenes*. The primers used for each of these genes are described in Table 1. PCR analysis involved a master mixture, which was composed of 10X Ex Taq Buffer, dNTP, MiliQ water, and Ex Taq DNA Polymerase from the TaKaRa Ex Taq™ DNA Kit (TaKaRa, Kusatsu, Japan), with the primer combination for each of the three genes previously mentioned. The amplification of the PCR products was performed in a DNA thermal cycler (BioRad, CA, USA) using the following program for all primer combinations: $94\ °C$ for 10 min (initial denaturation), followed by 40 cycles of [$94\ °C$ for 30 s (denaturation), $57\ °C$ for 30 s (annealing), and $72\ °C$ for 25 s (elongation)], and a final elongation at $72\ °C$ for 7 min. A sample containing only water was used as the negative control, whilst a sample containing the A4 plasmid was used as a positive control.

**Table 1.** List of primers targeting fragments of *rol*D, *vir*D, and 18S.

| Target | Orientation | Sequence | Amplicon Size | Reference |
|---|---|---|---|---|
| *rol*D | Forward | 5′-GCGAAGTGGATGTCTTTGGT-3′ | 225 bp | Lütken et al. [56] |
| | Reverse | 5′-TTGCGAGGTACACTGGACTG-3′ | | |
| *rol*D | Forward | 5′-CTGAATTACGACGCCTTGCG-3′ | 196 bp | Designed with Primer-BLAST using the accession X12867.1 position: 2855. |
| | Reverse | 5′-TGCGATGACGACTGTTCCAA-3′ | | |
| 18S | Forward | 5′-CGGCTACCACATCCAAGGAA-3′ | unknown | Topp et al. [57] |
| | Reverse | 5′-GCTGGAATTACCGCGGCT-3′ | | |

To visualize the PCR products, each sample was mixed with 6X GelRed® Prestain Loading Buffer with Orange Tracking Dye (Biotium, CA, USA), which contained 0.1% (*v/v*) Orange G tracking dye (ThermoFisher Scientific). The mixtures were loaded onto a 1.5% (*w/v*) agarose gel (VWR) and 3 uL of 100 bp DNA ladder (NEB, Quick-Load®, MA, USA). The PCR products were separated by running an agarose gel eletrophoresis at 120 V for 50 min and visualizing the result under UV-light (GelDoc, Universal Hood II, BioRad, CA, USA).

## 3. Results

*3.1. Transformation Efficiency of Different Inoculation Methods*

Overall, one HR was formed in the direct inoculation ex vitro method, that is, by directly injecting ex vitro seedlings with a liquid culture of *R. rhizogenes* (Table 2). However, no HRs were observed in the liquid co-culture ex vitro and dipping method, even though a large number of explants were used (900 explants were inoculated for liquid co-culture ex vitro). In the direct inoculation ex vitro method, only one of the seven inoculated seedlings formed HRs, which resulted in a natural transformation efficiency of 14.3%. In addition, two other methods were tested in order to induce and grow HRs in *Taxus* (Supplementary Files S1 and S2) with no efficiency.

**Table 2.** Transformation efficiency of different inoculation methods.

| Inoculation Method | Number of Inoculated Explants | Number of Hairy Roots Formed | Transformation Efficiency |
|---|---|---|---|
| Direct inoculation ex vitro | 7 | 1 | 14.3% |
| Liquid co-culture ex vitro | 900 | 0 | 0 |
| Dipping | 20 | 0 | 0 |
| Direct inoculation in vitro | 20 | 0 | 0 |
| Liquid co-culture in vitro | 18 | 0 | 0 |

### 3.2. Direct Inoculation Ex Vitro

When assessed after 2 months of inoculation, putative HRs near the inoculated site were found from the ex vitro seedlings inoculated by the direct injection of a liquid bacterial culture. Both the control and inoculated seedlings grew well under greenhouse conditions, with vegetative growth being observed (Figure 5A). The putative HRs emerged at a lower site than the infected site. The roots from both the control (Figure 5B) and inoculated (Figure 5C) seedlings were harvested for subsequent molecular analysis to confirm whether transformation was successful.

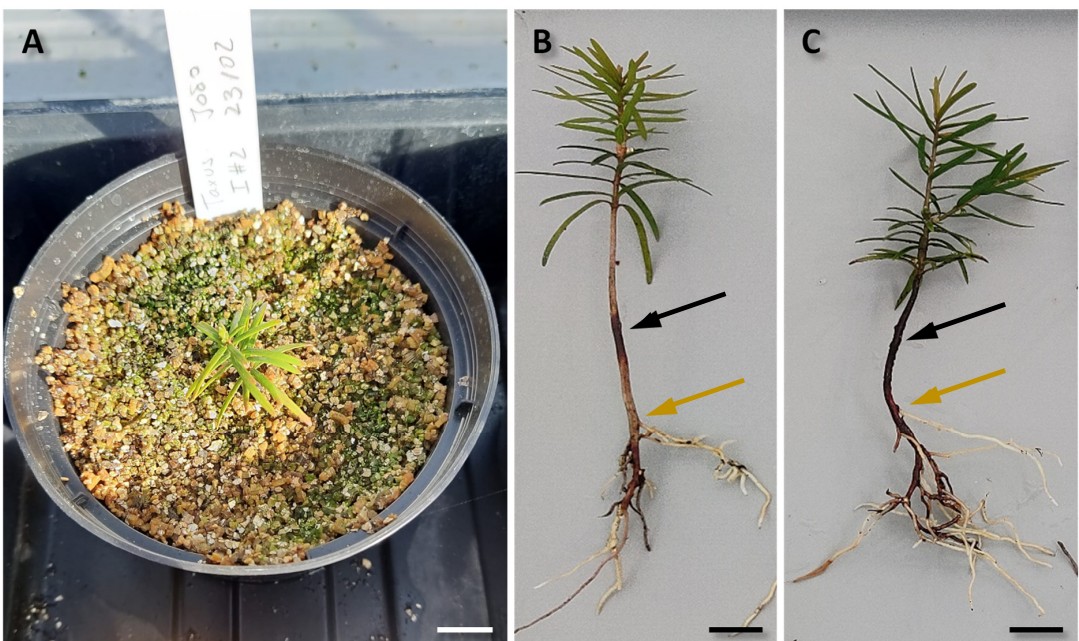

**Figure 5.** Seedlings inoculated by the direct inoculation ex vitro method after 2 months of inoculation. (**A**) Inoculated seedling grown in the greenhouse. (**B**) Control seedling. (**C**) Inoculated seedling; Black arrows indicate the inoculation site, while orange arrows indicate the highest protruding site of the formed roots. Bar = 1.5 cm for (**A**), and bars = 1.0 cm for (**B**,**C**).

### 3.3. Liquid Co-Culture Ex Vitro

The ex vitro needle inoculation experiment by liquid co-cultivation was severely affected by contamination. A total of 74% (data not shown) of the needles in the experiment showed contamination, which was seen as bacteria colonies with a yellow coloration surrounding the needles (Figure 6A). No HR formation was observed after one month of culturing the needles on rooting media and a severe discoloration/loss of greenness indicated tissue senescence, hence the experiment was terminated (Figure 6B).

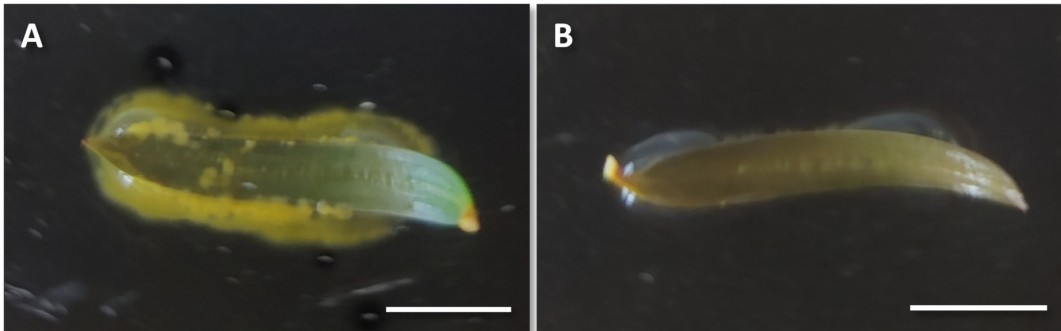

**Figure 6.** Needles inoculated by the liquid co-culture ex vitro method. (**A**) Contaminated needle. (**B**) Uncontaminated needle after 28 days of inoculation. Bars = 1.0 cm.

### 3.4. Dipping Method

The ex vitro shoots inoculated by the dipping method showed no HR formation after two months of being grown under greenhouse conditions (Figure 7). Moreover, the inoculated shoots developed a hard callus-like tissue at the site of inoculation (Figure 7A,B), which was not present on the control shoots (Figure 7C,D). As the shoots had not been sterilized prior to the inoculation, they suffered from fungal contamination as well as mealy bug infection.

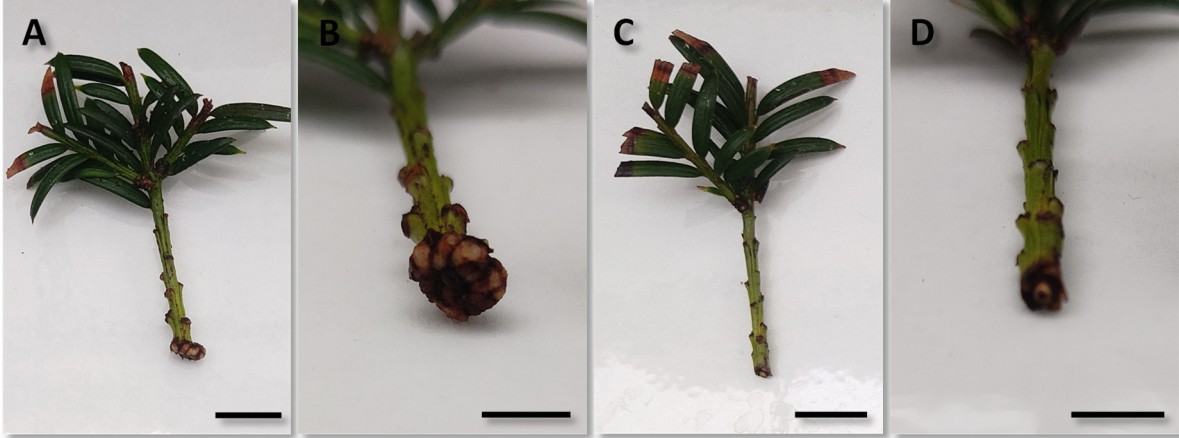

**Figure 7.** Ex vitro shoots inoculated by the dipping method after 2 months of inoculation. (**A**) Inoculated shoot. (**B**) Hard callus-like tissue on the inoculation site of the inoculated shoot. (**C**) Control shoot. (**D**) Inoculation site of the control shoot. Bars = 1.0 cm for (**A**,**C**), and bars = 0.5 cm for (**B**,**D**).

### 3.5. Direct Inoculation In Vitro and Liquid Co-Culture In Vitro (Supplementary Files S1 and S2)

No HR formation was observed from the in vitro shoots inoculated by directly smearing in a solid bacterial culture (Supplementary File S1). These shoots were inoculated and placed directly on the rooting media without a washing step, so signs of bacterial contamination were present since the beginning of the culture (data not shown).

The in vitro seedling inoculation by liquid co-culturing with bacteria (Supplementary File S2) showed no HR formation, even after four months of culturing. The inoculated seedlings developed callus tissue predominantly on the hypocotyl region (Figure 8A), unlike the control seedlings, which developed low or no formation of callus (Figure 8B).

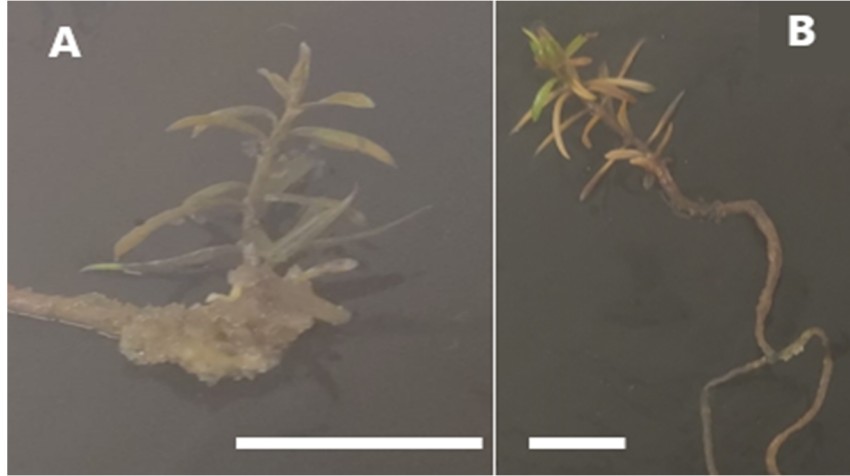

**Figure 8.** Seedlings inoculated by the liquid co-culture in vitro method after 28 days of inoculation. (**A**) Inoculated seedling. (**B**) Control seedling. Bars = 1.0 cm.

### 3.6. Molecular Analysis on the Putative HRs from Ex Vitro Seedlings

In the PCR analysis (Figure 9), presence of the *rol*D fragment in sample I2 was observed, while it was absent in the remaining samples and controls. The *vir*D fragment bands were not present in any root samples and controls. An 18S fragment was observed in samples I1, I2, I3, I5, I7, and C8. The A4 samples showed the presence of *rol*D, 18S, and *vir*D fragments, and none of these fragments in the N sample.

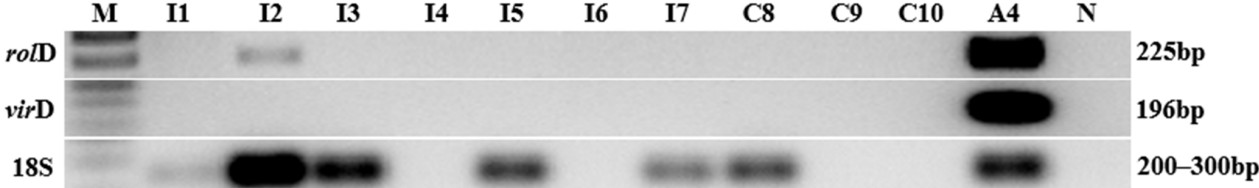

**Figure 9.** Molecular analysis from the root samples. PCR using *rol*D, *vir*D, and 18S primer sets; Length of the PCR products are shown; M: 100 bp DNA ladder; I1–7: root samples extracted from inoculated seedlings; C8–10: root samples extracted from control seedlings; A4: positive control with DNA from the A4 strain of *R. rhizogenes*; N: negative control with water.

The results confirmed that root harvested from sample I2 was in fact a HR induced by the natural transformation with *R. rhizogenes*, as it showed the presence of genes (*rol*D) from the T-DNA of the Ri-plasmid, whilst no signs of bacterial contamination were detected (no presence of *vir*D).

## 4. Discussion

For sustainable ecological and economic production of paclitaxel, HRCs, a rapidly developing biotechnology, was introduced to replace the direct extraction of paclitaxel from *T. baccata*, or the synthesis and semi-synthesis of paclitaxel [12–23]. Paclitaxel is produced by the induction of HRs generated from *T. baccata* with methyl jasmonate, which was reported on *Taxus* x *media* 'Hicksii', increasing the paclitaxel content by 3-fold in this way [48]. Therefore, it is pivotal to produce HRs through the transformation of *T. baccata* with *R. rhizogenes*.

In the current study, the A4 strain of *R. rhizogenes* was used for the inoculation. This strain has previously been pursued to transform *T. brevifolia* and *T. cuspidate* [45,46], although both attempts were unsuccessful. Since HRs from ex vitro seedlings were formed in the current study (Figure 5), it has been proven that the A4 strain can in fact transform *Taxus* spp., and more specifically, *T. baccata*, although the efficiency of the infection appears to be low. This is the first study describing the successful transformation of any *Taxus* spp. using the *R. rhizogenes* A4 strain.

During this study, HRs formed in the direct inoculation ex vitro method. More specifically, HRs were produced by directly injecting ex vitro seedlings of *T. baccata* with a liquid bacterial culture. Several other inoculation strategies were also investigated, involving other tissues such as shoots and needles as well as other inoculation methods such as the liquid co-culture ex vitro, dipping method, direct inoculation in vitro, and liquid co-culture in vitro. However, no HRs were observed. The direct inoculation ex vitro method successfully induced HRs with a transformation efficiency of 14.3% (Table 2). Direct infection methods have been used to successfully transform *T.* x *media* var. *hicksii* and *T. cuspidate* [45,48]. However, there are no reports of *T. baccata* tissue having been successfully transformed before the present work. Therefore, according to current knowledge, this is the first report of the successful transformation of *T. baccata* with *R. rhizogenes* that resulted in HR formation. Furthermore, there are no reports of *Taxus* spp. having been successfully transformed with *R. rhizogenes* in ex vitro conditions before, since natural transformation of *Taxus* spp. has always been conducted in vitro with plant material obtained from plants germinated by embryo rescue.

Different forms of the liquid co-culture method have been used over the years to induce HR formation in *T. brevifolia*, *T. cuspidata*, *T. sumatrana*, and *T. baccata* ssp. *Wallichiana* [45–47,50,58]. In this study, needles were inoculated by the liquid co-culture ex vitro method, although no HR formation was observed. The needles were subjected to a heat and centrifugation treatment during inoculation. This methodology was used by Sahai and Sinha [50], who obtained a 13% transformation efficiency. Centrifugation and heat treatments during infection with *A. tumefaciens* have been reported for embryogenic cell suspensions of banana (*Musa* spp.), and for immature embryos of *Oryza sativa* and *Zea mays* [53,54]. The protocol for the inoculation of needles by the liquid co-culture ex vitro method with the heat and centrifugation treatments was based on these two reports. Heat shock treatments are commonly used to make the tissue more permeable and susceptible to transformation. However, gravity-based treatments such as centrifugation are not as common and are less understood. Khanna et al. [54] proposed that centrifugation increased the contact between the suspended cells of banana and cells from *R. rhizogenes*. On the other hand, Hiei et al. [53] observed that centrifugation of rice embryos inhibited shoot elongation and promoted callus development, and therefore proposed that centrifugation negatively affects normal cell differentiation and organ development and promotes undifferentiated plant cell growth. It is more likely that the mechanism explained by the hypothesis of Hiei et al. [53] would be the one that would apply to the needles of *Taxus* spp., potentially increasing the transformation efficiency of the inoculation. Unfortunately, the needle cultures were extremely contaminated (Figure 6A). It was expected that the surface sterilization procedure would not be enough to completely sterilize the needle explants. Additionally, the endophytic fungi present in *Taxus* spp. are usually isolated after the plant samples are surface sterilized, indicating that surface sterilization is not enough to completely sterilize the explants [59,60].

The dipping method is unlike the liquid co-cultivation method in the sense that the explant is not completely submerged in the culture media. Although other studies such as Kim et al. [45] and Sahai and Sinha [50] have reported dipping methods for inoculation, these were not comparable to the dipping methods described during this study, as they simply consisted of submerging the explant for a short amount of time in the liquid culture of *R. rhizogenes*. For the dipping method in the present work, ex vitro shoots were the selected explant and were inoculated overnight by only submerging the shoot tip, ca. 1 cm. After inoculation, shoots were planted in pots and grown under greenhouse conditions similar to those of the ex vitro seedlings, which successfully formed HRs (Figure 7). As the shoots were not sterilized, signs of fungal and mealy bug contamination were detected in both cases. Nevertheless, callus still developed on the inoculated shoots (Figure 7B). Other studies showed that *Rubia cordifolia* formed callus after the transformation of *rol*B [61,62]. Similarly, Chen et al. [63] stated that calli were formed on hypocotyls of soybean inoculated with *R. rhizogenes*. In addition, HRs were not produced if the callus did not form [63], which was different from the present experiment because only the callus formed and no HR was subsequently generated.

The direct inoculation ex vitro method of seedlings described in the current work was based on the research of Nguyen and Searle [52]. However, liquid culture was used as an inoculum instead of a solid culture of *R. rhizogenes*. When assessed after 2 months of inoculation, HRs were found. The position of the root formation was different from the inoculation site (Figure 5). This might have been because a liquid culture was used as the inoculum, and therefore might be able to travel through the stem tissues after it injected with a syringe. The transformation was confirmed to have been successful for root sample I2, of which the presence of *rol*D was confirmed by PCR analysis (Figure 9).

Although the formation of HR was confirmed, no in vitro HR culture could be established because the experiment was performed ex vitro, and the seedlings were therefore unsterilized. Surface sterilization procedures would either destroy the root tissue, or be ineffective, most likely leading to endophytic fungi contamination after the roots were cultured in vitro. An alternative strategy would be to repeat the experiment using in vitro

germinated seedlings from embryo rescue (He, J. unpublished data), and use autoclaved peat and vermiculite as the substrate. If the transformation was again successful, there would be no need for complicated sterilization procedures before establishing the HRCs.

## 5. Conclusions

In the present study, HR formation of *T. baccata* was successfully achieved by natural transformation using the *R. rhizogenes* A4 strain, with a transformation efficiency of 14.3% generating one HR. This breakthrough finding revealed for the first time that the *R. rhizogenes* A4 strain is in fact capable of transforming *Taxus* spp., which paves the way for establishing a platform for producing paclitaxel. Additionally, it also closes the knowledge gap that *T. baccata* is susceptible to transformation with *R. rhizogenes*, which had not been previously reported.

**Supplementary Materials:** The following supporting information can be downloaded at: https://www.mdpi.com/article/10.3390/horticulturae9010004/s1, Supplemental File S1: Direct inoculation in vitro: in vitro shoot inoculation by directly inoculating with solid bacterial culture; Supplemental File S2: Liquid co-culture in vitro: in vitro seedling inoculation by liquid co-culture with bacteria.

**Author Contributions:** J.H.: Data curation, formal analysis, investigation, methodology, visualization, writing—original draft, writing—review & editing. J.P.A.P.: Data curation, formal analysis, investigation, methodology, visualization, writing—original draft, writing—review & editing. I.P.: Conceptualization, funding acquisition, project administration, writing—review & editing. S.K.: Conceptualization, funding acquisition, methodology, project administration, writing—review & editing. B.T.F.: Conceptualization, data curation, formal analysis, funding acquisition, investigation, methodology, supervision, visualization, writing—review & editing. H.L.: Conceptualization, formal analysis, funding acquisition, investigation, methodology, supervision, writing—review & editing. All authors have read and agreed to the published version of the manuscript.

**Funding:** JH reports the financial support provided by the China Scholarship Council (PhD grant no. 201906760024). BTF and HL report the financial support provided by Independent Research Fund Denmark Technology and Production (grant no. 0136-00410B).

**Institutional Review Board Statement:** Not applicable.

**Informed Consent Statement:** Not applicable.

**Data Availability Statement:** The data presented in this study are available on request from the corresponding author.

**Conflicts of Interest:** The authors declare no conflict of interest.

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
