# Peer review of "Hairy Root Induction of Taxus baccata L. by Natural Transformation with Rhizobium rhizogenes"

_horticulturae, doi:10.3390/horticulturae9010004_

Round 1

Reviewer 1 Report

Several minor changes should be performed on the manuscript. Please see the attachment. 

Reviewer 2 Report

Dear Authors,

The article in general is well written and the information presented in a logical manner but some information in sections Materials and Methods must be completed.

I have made several queries throughout the manuscript, most of them minor. For details, see the manuscript

Therefore, the present draft needs revision before further process.

Reviewer 3 Report

11- In Materials and Methods section, the authors should specify the origin of the strain R. rhizogenes A4.

22- In Table 1, primers for virD gen are not specified.

33- In Discussion section, the authors should mention roughly the way the production of T. baccata plants will be done to produce paclitaxel.
